# Novel and Efficient Methodology for Drop Placement Accuracy Testing of Robot-Guided Inkjet Printing onto 3D Objects

Robert Thalheim [1,*], Andreas Willert [1,*], Dana Mitra [1] and Ralf Zichner [1,2]

1 Fraunhofer Institute for Electronic Nano Systems ENAS, Technologie-Campus 3, 09126 Chemnitz, Germany; dana.mitra@enas.fraunhofer.de (D.M.); ralf.zichner@enas.fraunhofer.de (R.Z.)
2 Faculty of Electrical Engineering, University of Technology Chemnitz, 09107 Chemnitz, Germany
* Correspondence: robert.thalheim@enas.fraunhofer.de (R.T.); andreas.willert@enas.fraunhofer.de (A.W.)

**Abstract:** Robot-guided inkjet printing technology offers a new way for the digital and additive deposition of low-viscous inks to be made directly onto arbitrary surfaces and, thus, enables the production of individualized printed electronics on large-scale objects. When compared to conventional flatbed printing, the distance between the nozzle plate and the object's surface varies and needs to be considered in order to match the accuracy requirements needed for the positioning of single drops. Knowledge about applicable distance limits and the influence of tunable print parameters is crucial for improving the print process and results. This study discusses the sources of errors in the inkjet printing process onto 3D objects and presents extensive results about position accuracy in relation to jetting distance for different parameter sets of functional inks, drop volumes, and piezo voltages. Additionally, an efficient novel method was applied to determine the drop position accuracy of inkjet droplets in relation to the jetting distance. The method relies on cylinder geometry for the object and an inkjet head that is guided by a six-axis robot manipulator along the cylinder's axis. For the determination of drop placement accuracy, the position of single dots on the surface was compared to a model which considered the cylinder radii, drop velocity, and the movement speed of the guided inkjet printhead. The method and the extensive research results can be utilized for the prediction of achievable drop placement accuracy and the prior definition of distance limits.

**Keywords:** inkjet; inkjet printing; robot; accuracy; placement; jetting distance; 3D object





## 1. Introduction

Inkjet printing is a versatile deposition technology widely utilized today [1–3]. Originally developed for graphic arts [4], the advantages of digital imaging moved quickly into the textiles industry [5] and into the decoration of ceramic tiles [6] and laminate flooring [7]. In the meantime, inkjet printing has received huge attention for manifold electronic and functional applications, with the number of publications made in the last 20 years impressively reaching about 20,000 contributions (Google Scholar search from 13.04.2023; the search term "functional inkjet printing" delivered 19,700 results [time period: 2003–2023; excluding citations]). The scope lies in between optical, electrical, chemical, biological, and 3D printing [8–10] applications. Functional materials, which are used for inkjet printing, provide, for instance, electrical conductivity, semi-conductivity, insulation, piezoelectricity, luminescence, or chemical sensing to create, e.g., printed antennas [11], sensors [12], capacitors [13] or transistors [14]. Most commonly, the printing substrate is flat, and the printing direction is upside down. The inkjet head is either fixed while the substrate moves, or vice versa.

A new degree of freedom is generated when an inkjet head is combined with xyz positioning or even a robot system. This enables the contact and maskless additive deposition of functional materials directly onto an object. By using standard industrial robots, a guided inkjet printhead can perform motions in any direction in 3D space and can be applied in

production facilities, such as in the automotive or aviation industries. Regarding future prospects, this enables the digital and automatized production of electronic components, like parts of a wiring harness, sensors, heating elements, or antennas, to be used in wireless communication on the large-scale surfaces of cars and airplanes.

The first described systems, which were composed of an inkjet printhead and a robot multi-axis system, were meant to deposit decoration colors onto vehicles or even airplanes. Jean-Pierre Gazeau et al. reported on the development of a five-axis inkjet printing robot for the printing of images onto 3D, wide area surfaces in a publication from 2009 [15] and a patent from 2006 [16]. Further applications saw the decoration of shoes, bottles, or balls. Many patents have been filed for this method and for different technical improvements regarding robot-guided printing technology to colorful printing [17–19]. The challenges in these kinds of applications are the generation of the inkjet head pathway as well as the stitching of several swathes to generate colorful images without any missing parts. The first known publication, which addresses the topic of "functional inkjet printing on 3D", was published in 2005 [20]. Scientists used a continuous inkjet printhead and a 2D motion system to print an antenna structure onto a glass cup. Only a few further scientific publications are known that focus on functional inkjet printing onto objects in combination with robot systems [21–25]. In these publications, the setups are composed of an inkjet printhead that was mounted onto a six-axis robot manipulator. Therein, the results and properties of electrical applications were demonstrated, like conductive tracks on a microscope glass slide [21,22], a capacitive level sensor [23,24], and basic research on the flow behavior of printed conductive layers on inclined planes, including inline infrared (IR) post-treatment to facilitate electrical resistance homogeneity [25].

When compared to conventional functional inkjet printing on flat surfaces, major differences, and additional error sources have to be considered regarding the robot-guided inkjet approach. Firstly, the printing direction is not solely upside down but also angled with respect to the gravitational field. Additionally, the printhead, which is a 2D array consisting of hundreds of nozzles, moves over a curved 3D surface. Therefore, its throw distance is increased and differs from one nozzle to another, which has more impact. Finally, the printhead assembly has to work with a certain additional standoff to avoid a collision, especially when printing into cavities. Knowledge of the applicable process parameters, like the maximum jetting distance, is key to achieving the desired accuracy and, in the end, proper functionality of the printed functional layers.

The current investigations discussed in this paper represent fundamental experiments for understanding inkjet basics and determining the dot placement accuracy that is achievable in robot-controlled inkjet printing with respect to three-dimensional objects. The inkjet printing process was investigated down to the drop size level to understand and determine the process limits for the maximum applicable jetting distance in inkjet printing onto 3D objects. Therefore, comprehensive printing experiments were conducted on different cylinders for a solvent-based conductive silver nanoparticle ink and a dielectric solvent-free UV curable ink, as well as for different drop sizes and driving voltages. The print results were analyzed and compared to a specifically developed model, which calculates the ideal position without the deflection of single droplets on the cylinder surface according to the print configuration. Finally, out of this comparison, the placement error in relation to surface distance was determined for all parameter sets. By the reversal calculation of fitted functions, a given placement tolerance can be used to determine the maximum applicable jetting distance. Along with knowledge about the influence of the jetting distance and projection-based geometrical deviations, the robotic printing process can be improved by defining limits that can be applied to the printing process.

In the following Section 2, we first give an overview of the error sources for placement errors in inkjet printing onto arbitrary surfaces. Following this, the robot printing system is described in Section 3. In Section 4, the basic experiments and the mathematical model are presented. Finally, the results are discussed in Section 5, and in Section 6, our conclusions are drawn, and future perspectives are given.

## 2. Sources of Error for Inkjet Drop Placement on Curved Surfaces

Conventionally, inkjet printing is performed on flat substrates in a roll-to-roll or sheet-to-sheet process where the printhead's nozzle plate and the substrate are in parallel, and the distance can be easily adjusted to a few millimeters. In contrast, for inkjet printing onto a curved, 3D object, a number of additional factors and effects will influence the quality of the inkjet-printed pattern. In Figure 1, a systematic overview of the possible origins of the deviation from the intended drop placement of the printed ink is given regarding the robot-guided inkjet printing process. Hereinafter, these error sources are briefly discussed and are distinguished into three main groups.

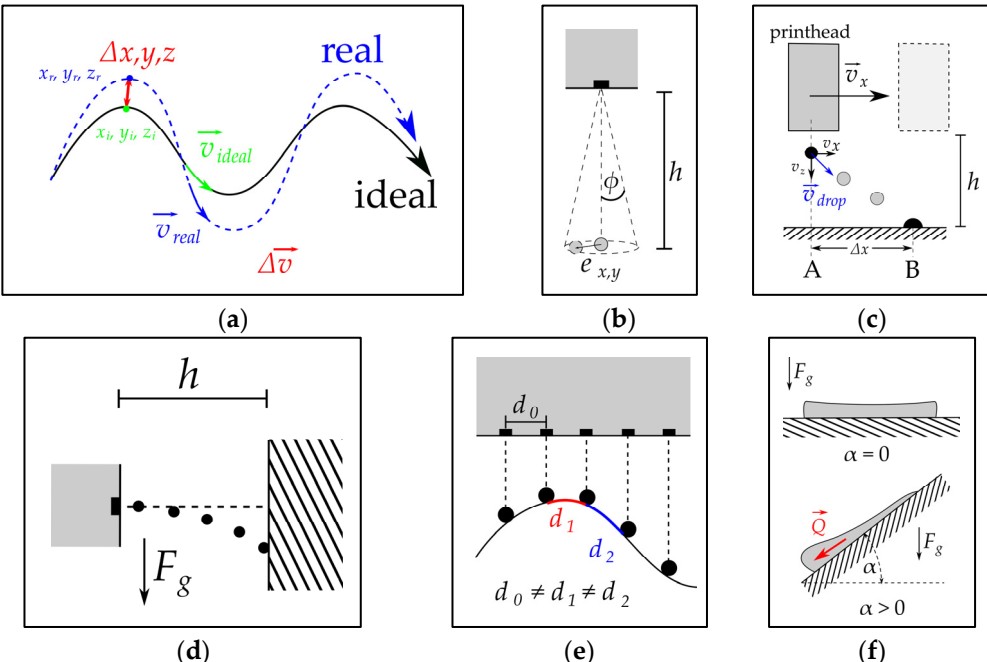

**Figure 1.** Sources of placement errors for the robot-guided inkjet printing process: (**a**) robot motion along a trajectory (see Section 2.1); (**b**) jetting straightness (see Section 2.2.1); (**c**) drop motion in print direction (see Section 2.2.2); (**d**) printhead spatial orientation (see Section 2.2.3); (**e**) distortion by geometrical projection (see Section 2.3.1); (**f**) liquid flow behavior (see Section 2.3.2).

### 2.1. Robot Motion, Path Planning, and Scanning System

In the first place, the motion system (e.g., a six-axis industrial robot manipulator) limits the achievable accuracy of how an inkjet printhead can be guided along a surface. The printhead's real trajectory $(x_r, y_r, z_r)$ as well as its velocity, $\vec{v}_{real}$, along the path, underlie the intrinsic deviations $(\Delta x,y,z, \Delta \vec{v})$ from the ideal path $(x_i, y_i, z_i)$ and velocity, $\vec{v}_{ideal}$, (as shown in Figure 1a) caused by the complex combined motion of multiple linear and/or rotational axes. When considering that a prior scanning process is needed to detect the shape, position, and orientation of a workpiece, we see that discrepancies between the digitized and the real object contribute to the overall error. In terms of large-scale patterns, which are composed of multiple swaths, each track has to match the previous one, and the robot needs to guide the inkjet printhead accordingly, always including inevitable imprecision. Therefore, the processes of scanning, path planning, and moving the printhead via a robot manipulator will contribute to placement errors.

### 2.2. Inkjet Printing Process

Besides the described motion system-related issues, the inkjet printing technology itself is another major source for placement deviations regarding the droplets. Three main relationships are found to be responsible for the majority of dot placement errors.

### 2.2.1. Jet Straightness in Relation to Print Distance

In Figure 1b, a placement error $e_{x,y}$ from the intended position is shown. These kinds of errors are caused by the imperfection of the printhead's jet straightness, which is usually quantified by the angle $\phi$. Obviously, these errors are related to the travel distance $h$ of the single drops. Additionally, the speed of inkjet droplets is continuously slowed down on their way to the substrate, and the influence of air turbulences will contribute to further displacement. Therefore, it can be stated: the longer the jetting distance $h$ between the nozzle position and the object surface, the higher the resulting displacement $e_{x,y}$ becomes. Therefore, a major strategy for inkjet printing is to minimize the distance between the printhead and the surface. By taking into account the printhead nozzle plate's 2D array-like geometry in contrast to the shape of a three-dimensional object, the required maximum distance might be exceeded, especially when it comes to printing into cavities or onto convex surfaces with small radii.

### 2.2.2. Drop Motion in Print Direction

Suppose a printhead is moved in the x-direction across a surface with an offset, $h$, and a certain speed, $\vec{v}_x$. When the drops are ejected, the velocity, $\vec{v}_{drop}$, is then composed of the velocity, $\vec{v}_z$, towards the surface and the motion velocity, $\vec{v}_x$, of the printhead. The small droplets are slowed down constantly by aerodynamic drag and buoyancy and will pass the gap within a certain time. This causes displacement, $\Delta x$, between the initial printhead position A (drop creation) and the impact position B and is related to the distance $h$ and the development of $\vec{v}_{drop}$ (Figure 1c). For conventional printing onto two-dimensional surfaces, all drops will have the same travel distance until they impact the substrate. When it comes to printing onto non-flat objects, the time of flight and, consequently, $\Delta x$ differ, causing an unequal distribution that has to be considered.

### 2.2.3. Printhead Spatial Orientation

By taking into account the fact that, in a robot-guided inkjet printing process, the ejection of drops might occur in any spatial orientation, we see that the force of gravity, $F_g$, might also influence placement accuracy (see Figure 1d). Assuming the printing occurs with an angle of 90° (relevant to gravity), the drops will be deflected in the direction of the force of gravity. The deflection is dependent on the distance, $h$, mass, speed, and size/shape of the drop.

### *2.3. Layer Formation*

For the functional layers, shape and homogeneity are crucial for performance (e.g., electrical conductivity). In view of the printing of functional layers onto 3D surfaces, additional error sources can be identified. Hereafter, two major causes, which provoke irregularities or malfunctions, are described.

### 2.3.1. Distortion by Geometrical Projection

Inkjet printing onto non-planar surfaces will lead to the additional geometrical displacement of droplets caused by the projection of a 1- or 2D array of nozzles (the printhead) onto a curved surface (see Figure 1e). According to the topography of the object's surface, the distance, $d_i$, in between two drops on the surface is not equal ($d_0 \neq d_1 \neq d_2$). In graphic arts image printing, isolated single dots will generate an image impression for a human observer. A displacement of inkjet droplets might cause moiré or other visual effects, like image distortion. In contrast, for functional inkjet printing, coherent and uniform material distribution is necessary for the realization of the desired function (e.g., electrical conductivity or insulation) and to achieve drop coalescence. The distortion caused by the projection might lead to an uneven layer thickness or missing material and, thus, to a malfunction.

2.3.2. Liquid Flow Behavior

The printing of functional liquid patterns onto the unlevel regions of a 3D object might result in inhomogeneous layer formation. As illustrated in Figure 1f, the force of gravity, $F_g$, will lead to a flow, $\vec{Q}$, of a liquid film, depending on the inclination angle, $\alpha$. Furthermore, the surface (surface free energy and roughness), ink properties (viscosity and surface tension), and layer morphology (pattern size and thickness) influence the development of flow [25].

In summary, all the factors described in Section 2 have to be considered. It is crucial that the position of the drops at the surface is highly accurate and preferably equidistant to achieve the best prerequisites for layer formation. In reality, this requirement cannot be entirely fulfilled due to the discussed sources of errors. The maximum tolerable displacement has to be considered for each application with respect to the geometrical and electrical properties needed (e.g., feature size, linewidth, pitch, conductance, and layer homogeneity). This study focuses on the influencing factor of the height, $h$, with regard to the placement accuracy of single drops within the printing process in which the printhead moves. A novel and efficient method for the determination of the placement error in relation to the distance is presented.

**3. Experimental Setup and Methodology**

The experiments were conducted using a setup (see Figure 2) based on the six-axis robot arm GP8 controlled by the YRC1000 robot controller from Yaskawa (Figure 2a). The manipulator's repeatability is ±10 µm, with a maximum payload of 8 kg and a maximum working range (exclusive tool) of 727 mm, according to manufacturer specifications. At the flange, a mounting frame is attached, whereby different tools can be installed. For the printing experiments, the robot was guiding one out of three Q-Class piezo inkjet printheads from Fujifilm Dimatix (Figure 2b), which consist of 256 square-shaped nozzles in one row, with a native resolution of 100 dpi (i.e., a distance of 254 µm between two nozzles), and a calibrated drop volume of 10 pL, 30 pL, or 80 pL, respectively. The diameters of the orifices are 31 µm (10 pL), 42 µm (30 pL), and approx. 60 µm (80 pL). The printhead controller (Mercury Development Kit from Fujifilm Dimatix) enables the user to drive the signal—also called a waveform—which is applied so as to electrically drive the piezo crystals. Besides the signal shape (rise, hold, and fall time), the operating piezo voltage and frequency can be altered, as well as the temperature within the printhead assembly (increasing temperature will lower the ink's viscosity). Furthermore, a meniscus pressure pump was used to create a negative pressure within the ink reservoir in the range of 6–9 mbar to prevent the ink from leaking. The value was adjusted depending on the ink properties and level.

Two different inks were used for the experiments:

1. AGF—a black UV-curable ink from Agfa (Altamira Pack LMX) with a density of 1.09 g/cm$^3$, 9–11 mPa·s viscosity (T = 45 °C), and 22.5 mN/m ± 1 mN/m (T = 25 °C) surface tension;
2. PVN—a silver nanoparticle ink from PV Nanocell (I40DM-106) with a density of 1.62 g/cm$^3$, 10 mPa·s viscosity (T = 25 °C), and a silver load of 40%. The surface tension was not determined.

For the initial determination of drop volume and the velocity for both inks, the drop-watching device jetXpert OEM (ImageXperts Inc., Nashua, USA) was used (Figure 2c). Flexible photo papers with a nanoporous layer were used as substrates mounted directly on the cylinder surfaces (three different radii: 35 mm, 45 mm, and 55 mm) for the printing experiments. Microscopic images from the print results were taken with a microscope from Zeiss (Axio Imager M2). The whole image width (up to 83 mm) was covered by multiple images, which were stitched. For image processing, the ImageJ software (developed by the National Institute of Mental Health (NIMH)) was employed. The software is available on the RSB home page (https://imagej.nih.gov/ij/index.html, accessed on 10 February

2023). The evaluation of the displacement error was conducted by using a program that was written in Matlab (The Mathworks, Inc., Rev. R2021b).

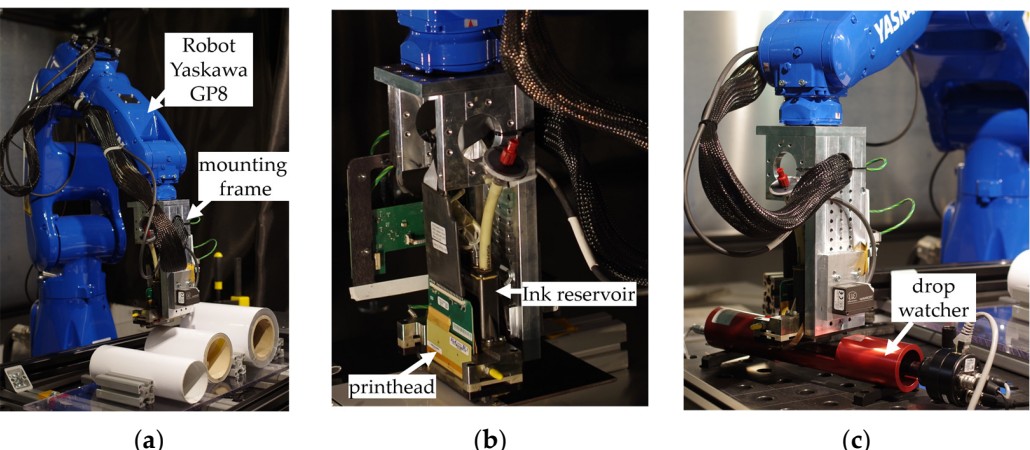

| (a) | (b) | (c) |

**Figure 2.** Overview of experimental setup: (**a**) robot printing system: Yaskawa GP8 in position for printing onto cylinders; (**b**) close view end effector: mounting frame with attached printhead Dimatix Sapphire QS; (**c**) printhead in position for drop watching experiment.

## 4. Experiments and Modeling

The focus of the experiments was related to the displacement errors resulting from the surface-to-printhead distance, as described in Section 2.2.1—Jet Straightness in Relation to Print Distance. Therefore, the printing process was investigated down to the drop size scale in two sets of experiments: In Section 4.1—Drop Watching, the basic parameters of drop size and speed for different kinds of setups are reported. In Section 4.2—Printing onto a Cylinder, the placement accuracy in relation to the jetting distance is determined using a method based on a comparison between the model and the experimentally observed drop positions on the cylinder surface. The influence of gravitational force, which resulted in a liquid flow within the isolated droplets used in the experiments, was assumed to be neglectable. Within this study, in total, 36 combinations of ink, drop volume, cylinder radius, and voltage were investigated.

### 4.1. Drop Watching

The prior characterization of inkjet droplets is essential for drop trajectory modeling and the calculation of the placement error. The nominal drop volume of each printhead is related to the nozzle geometry and its size. These values are specified by the printhead manufacturer. The actual ejected drop volume might differ and depend on certain parameters, like driving voltage, waveform, ink viscosity, and surface tension. For the current experiments, we have chosen printheads (Fujifilm Dimatix Q-Class Sapphire) with nominal volumes of 10 pL, 30 pL, and 80 pL, respectively, and tested two different inks. Additionally, three different driving voltages were used: 80 V, 100 V, and 120 V. The waveform was fixed for each parameter set of printheads, inks, and piezo voltages.

The distance, $h$, between the nozzle plate and observation spot varied between 300 µm and 8400 µm, depending on the setup. At each stage, the drop velocity and volume were determined. By using the regression curve for the velocity values, $v_z(h)$, at different distances, $h$, according to Equation (1), $v_{z,0}$ and the slope, $m$, were determined.

$$v_z(h) = v_{z,0} + m * h \qquad (1)$$

The results of these experiments were used for the drop position model for printing onto a cylinder.

### 4.2. Printing onto a Cylinder and Modeling

4.2.1. Print Setup

For the investigation of the drop placement accuracy in relation to the jetting distance, a setup was deployed with a printhead above a cylinder. The printhead was oriented perpendicular to the cylinder axis, and its motion was directed along this axis (see Figure 3a,b). The center of the printhead was aligned with the axis of this cylinder. In this setup, the center of the printhead has the smallest distance to the cylinder surface while the outermost nozzles have the largest. For these experiments, cylindrical plastic tubes with three different radii $r_a$ ($r_{a,1}$ = 55 mm, $r_{a,2}$ = 45 mm, and $r_{a,3}$ = 35 mm) were covered with photo paper. The robot was programmed to guide the printhead in a linear motion with a constant speed of 152 cm/min and constant minimum distance of approx. 1 mm between the center nozzle and the cylinder surface (Figure 3c). The fire frequency of the printhead was 0.1 kHz (i.e., a distance of 253 μm between two droplets in the printing direction and of 254 μm across the printing direction). The printing was performed using both inks and different native drop volumes (PVN = 10 pL and 30 pL; AGF = 30 pL and 80 pL) and three different driving voltages (80 V, 100 V, and 120 V). Following the printing process, no post-treatment (temperature, UV) was applied. The samples were dried in a laboratory environment.

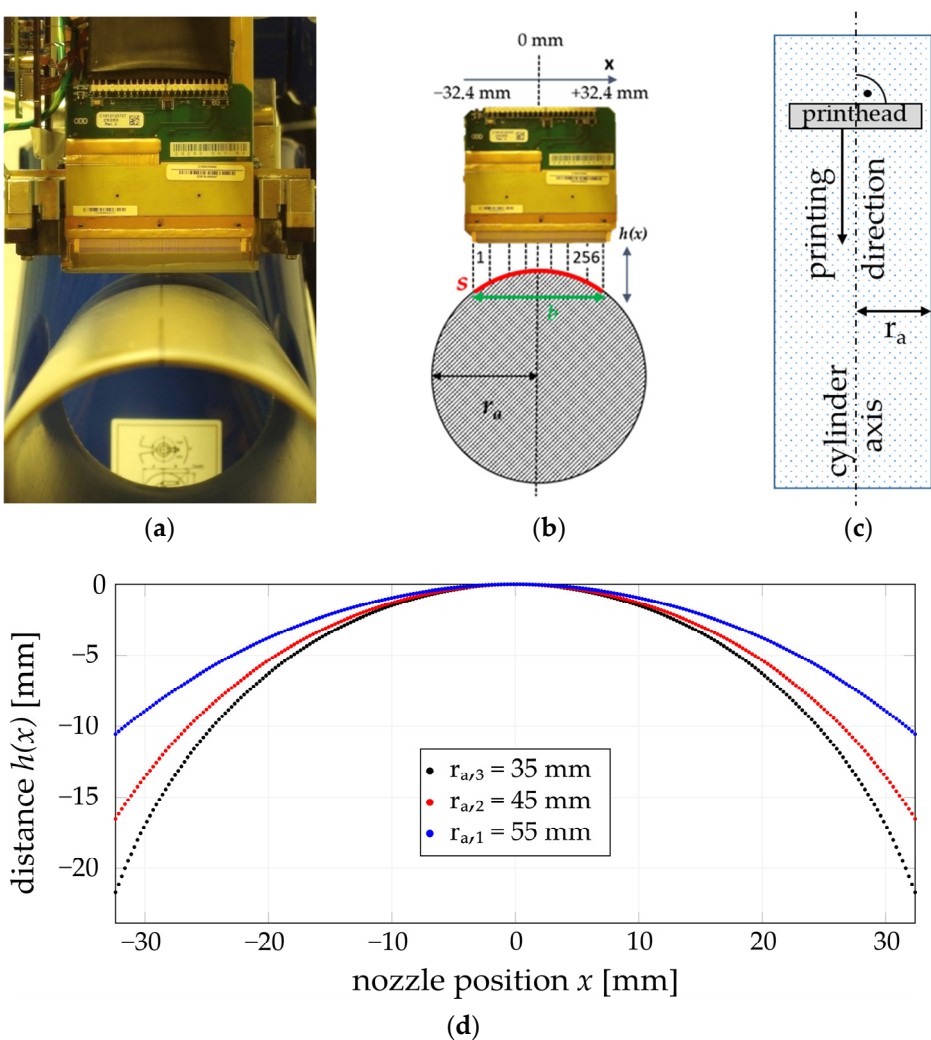

**Figure 3.** Printing process onto cylinder geometry for the determination of drop placement accuracy: (**a**) printhead in position for printing onto a cylinder with $r_{a,1}$ = 55 mm; (**b**) visualization of the projection onto a cylinder surface (true to scale for $r_{a,1}$ = 55 mm); (**c**) top view of print direction; (**d**) diagram of distance, $h(x)$, in relation to the nozzle position, $x$ (Equation (2)).

### 4.2.2. Modeling

Following the print setup description, the mathematical modeling of printing onto the cylinders was performed.

#### Print Distance

In relation to the cylinder radius, $r_{a,i}$, and the printhead nozzle (Figure 3b) at position $x$, the distance, $h_i(x)$, towards the cylinder surface can be calculated using Equation (2) based on the Pythagorean theorem:

$$h_i(x) = r_{a,i} - \sqrt{r_{a,i}^2 - x^2} \tag{2}$$

For $r_{a,1} = 55$ mm, the distance, $h_1(x)$, of the outermost nozzle at $x = \pm 32.385$ mm is approx. $h_1(\pm 32.385) = -10.5$ mm. At $r_{a,3} = 35$ mm, $h_3(x)$ is nearly $h_3(\pm 32.385) = -21.7$ mm, as shown in the diagram in Figure 3d. Assuming the jetting occurs in an ideal, static pose without any deviation, a simple projection occurs and reveals the ideal position of the resulting dots on the surface.

#### Deviation in Relation to Print and Drop Speed

In a dynamic printing process, both motion, $v_y$, and ejection, $v_z$, speed will lead to a distorted print image regarding this cylinder setup (see Section 2.2). Equation (3) describes the deviation, $\Delta y$, in print respective to the $y$ motion direction.

$$\Delta y_i = \frac{h_i\, v_y}{v_z} \tag{3}$$

According to [26], the drop velocity, $v_{drop}$, is constantly reduced by the drag force and buoyancy. Moreover, the drop volume, cross-section, and density, as well as the drag coefficient, determine the reduction in speed and the maximum jetting distance. Instead of using a numerical approach to calculate the drop velocity, $v_z$, our model relies on empirical data from the observation of the drop flight via a drop watcher in different height positions according to Equation (1) (see Section 4.1—Drop Watching).

In relation to the distance, $h_i$, between the nozzle plate and the surface, the average velocity, $\overline{v_z}$ (Equation (4)), was used for the calculation of $\Delta y,i$. For the model, the movement speed, $v_y$, was assumed to be constant.

$$\overline{v_z} = v_{z,0} - \left[ \frac{v_{z,0} - v_{z(h_i)}}{2} \right] \tag{4}$$

For assessing the print quality, we obviously needed to compare the ideal projection with the real position of single dots on the printout. For the calculation of the ideal projection, it was necessary to consider line elongation according to the cylinder radius.

#### Image Distortion

In relation to $r_{a,i}$, the overall length, $s_i$, of the arc (see Figure 4a) between the first (#1) and last dot (#256) on the cylinder surface is calculated using Equation (5).

$$s_i = 2 * r_{a,i} * \sin^{-1}\left( \frac{b}{2 r_{a,i}} \right); \qquad b \leq 2 * r_{a,i} \tag{5}$$

The print width, $b$, for a Q-Class printhead, is 64.77 mm. According to Equation (5), the print image width is elongated to a total arc length of $s_1 = 69.26$ mm ($r_{a,1} = 55$ mm), $s_2 = 75.48$ mm ($r_{a,2} = 45$ mm), and $s_3 = 82.73$ mm ($r_{a,3} = 35$ mm), respectively.

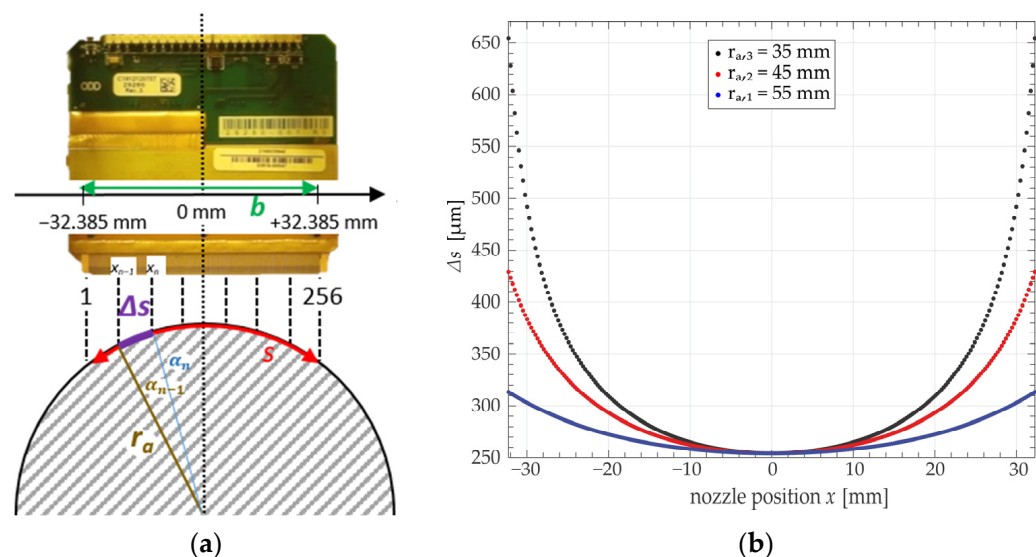

**Figure 4.** Illustration of image distortion: (**a**) projection of the nozzle distance at the cylinder surface; (**b**) diagram: arc distance, $\Delta s$, as a function of the nozzle position, $x$ (Equation (6)).

The distribution of a grid of dots resulting from the ideal projection on the cylinder surface should result in an equal distance between the subsequent dots in the print direction (at constant print speed and frequency) and an increase in distance, $\Delta s_{n,i}$, across the print direction. According to the position of a nozzle, $x_n$ ($2 < n \le 256$), with regard to the position of its next neighbor, $x_{n-1}$, the distance, $\Delta s_{n,i}$, is calculated as follows:

$$\Delta s_{n,i} = r_{a,i}\left(\left|\alpha_{x_n} - \alpha_{x_{n-1}}\right|\right) = r_{a,i}\left(\left|\sin^{-1}\left(\frac{x_n}{r_{a,i}}\right) - \sin^{-1}\left(\frac{x_{n-1}}{r_{a,i}}\right)\right|\right) \tag{6}$$

In relation to the radius, the length, $\Delta s_{n,i}$, will increase on the cylinder surface the further away the position, $x_n$ and $x_{n-1}$, of the nozzles is located from the center point. The impact locations of two neighboring drops on the cylinder surface are correlated to the angles $\alpha_{x_n}$ and $\alpha_{x_{n-1}}$ (see Figure 4a). The results of Equation (6) are shown in Figure 4b. The minimum value is 254 µm, which is the distance of the nozzles, according to a print resolution of 100 dpi.

Summarized Model Equation

By combining Equations (1)–(6), the calculation of the model position, $x_{cylinder,i}$ and $y_{cylinder,i}$, on the cylinder surface is summarized by Equations (7) and (8). In Equation (7), the lateral position, $x_{cylinder,i}$, is dependent on the printhead nozzle position, $x_{printhead,i}$, and the cylinder radius, $r_{a,i}$. The y position, $y_{cylinder,i}$ (Equation (8)), equals the deviation, $\Delta y$, from the ejection origin with respect to the print motion speed, $v_x$, and initial drop ejection speed, $v_{z,0}$. The coefficient $c$ represents the intercept of the linear equation for drop velocity development and can differ from the true ejection drop speed, $v_{z,0}$, at the nozzle surface when only the linear aspect of drop speed development is considered for the linear regression.

$$x_{cylinder,i} = r_{a,i}\sin^{-1}\frac{x_{printhead,i}}{r_{a,i}} \tag{7}$$

$$y_{cylinder,i} = \Delta y = 2 * \frac{v_x * \left[r_{a,i} - \sqrt{r_{a,i}^2 - x_{printhead,i}^2}\right]}{v_{z,0} + m * \left[r_{a,i} - \sqrt{r_{a,i}^2 - x_{printhead,i}^2}\right] + c} \tag{8}$$

### 4.2.3. Determination of Displacement and Maximum Height

The printing results were analyzed through stitched microscopic images covering the full print area. Based on image processing using ImageJ, the $x_d$ and $y_d$ coordinates of single dots within the digital image were detected by a procedure. Firstly, threshold binarization was applied, following an analyzing step, to determine the center of mass (CM) of the individual drops. One image usually covers nine printed lines in the print direction and a maximum of 2304 dots. A Matlab program (see supporting information matlab files) was developed to compare the experimental dataset with the model dataset, which was calculated based on empirical data from the drop watching experiments. Prior to further processing, an iterative, closest point (ICP) algorithm [27] was applied to converge both datasets. Within the following step, the drop positions from the experimental dataset were compared to the positions calculated in the model dataset, according to the k-nearest neighbor algorithm, to determine a pair of associated dots. The position of each pair, along with information about the distance between the printhead and surface, was then used to calculate the displacement error in the print direction in relation to the travel distance. Furthermore, an exponential function was fitted to the averaged values of displacement in relation to height, according to the following equation:

$$\Delta y = a \cdot e^{f h_i} \tag{9}$$

The displacement, $\Delta y$, is dependent on the distance, $h_i$, and the regression coefficients $a$ and $f$. By rearranging Equation (9), the maximum distance can be calculated in relation to deviation $\Delta y$ using Equation (10).

$$h_{max,i} = \frac{1}{f} ln \left( \frac{\Delta y}{a} \right) \tag{10}$$

## 5. Results and Discussion

In this chapter, the results and important findings from the extensive experiments are given and discussed in a condensed manner. Further detailed data and figures (PDF1-3) can be found in the Supplementary Material.

### 5.1. Drop Watching

The purpose of the drop watching experiments was to determine the initial droplet velocity and the following droplet movement. In Figure 5, exemplary images from the drop watching experiments are shown, which were used for the measurement of drop speed and volume. The set of images in Figure 5a,b show the drop ejection at the initial state at a delay ($t_{delay}$) of 40 µs, respectively 50 µs, between piezo contraction and image acquisition. For both inks (Figure 5a: PVN and Figure 5b: AGF), the increased piezo voltage led to a more distant position in the image section. The sequence of images in Figure 5c shows the drop formation of the AGF ink ejected out of an 80 pL print head, which was driven by 80 V piezo voltage. The images were taken at different moments, which is indicated in the lower row of the image. Additionally, the distance between the main drop to the nozzle plate is given. The tail of the jet is present until 200 µs and falls apart in between 200 and 300 µs. In the following images, at up to 500 µs, at least one satellite drop remains. This image sequence shows the typical formation of inkjet droplets. For other combinations at a higher voltage level (e.g., PVN, 10 pL, and 120 V), the number of satellites increased. For the ink under 80 V of piezo voltage for drop speed and volume, only the main drop was counted.

A summary of all the investigated combinations is given in Table 1, containing the average measured drop volume as well as the coefficients from the linear regression (Equation (1)). These results were used for the modeling of droplet position on a cylinder surface. The measurement results for drop velocity for the QS 10 pL printhead ejecting the silver ink PVN are plotted in a diagram in Figure 6 as an example. Additionally, Figure 7 compares the drop velocity of the tested inks and the drop volumes in relation to the distance, $h$, at 80 V driving voltage.

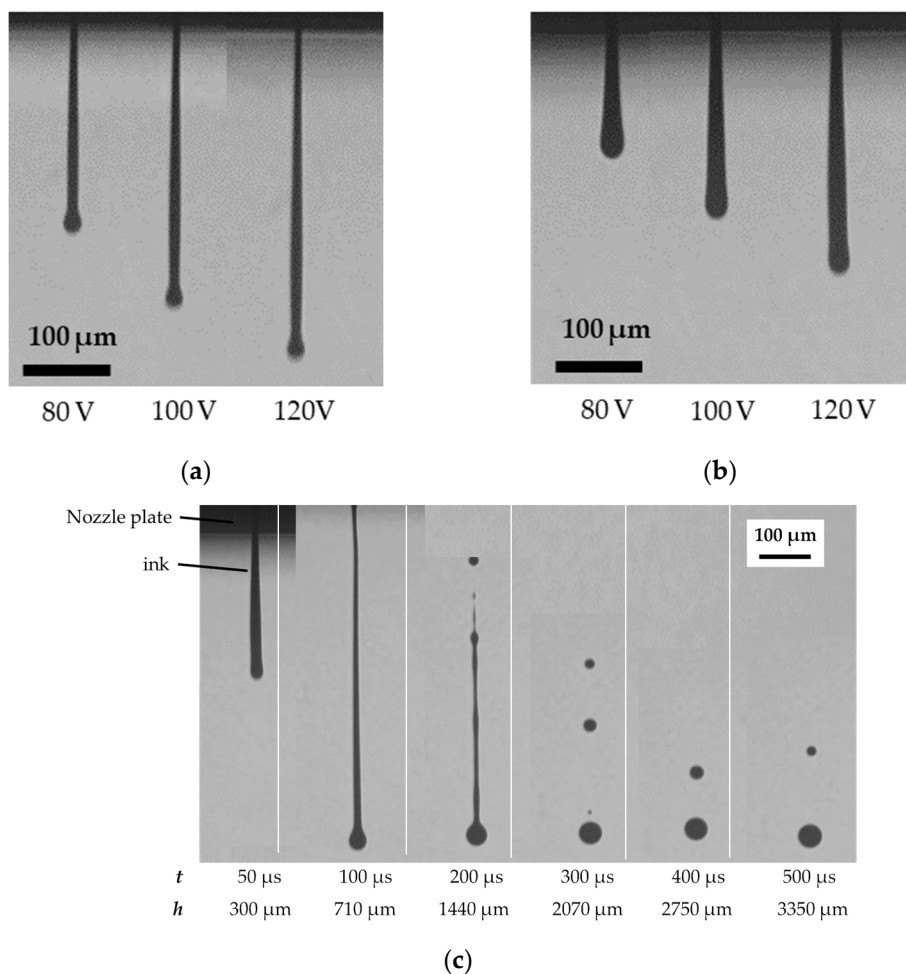

**Figure 5.** Exemplary images for drop watching experiments: (**a**) PVN (@ 30 pL nom. drop vol., $t_{delay}$ = 40 µs) and (**b**) AGF (@ 80 pL nom. drop vol., $t_{delay}$ = 50 µs) for different voltages; (**c**) drop formation in relation to time, $t_{delay}$, and main drop distance, $h$, to nozzle plate for 30 pL nom. drop volume for AGF.

**Table 1.** Coefficients, $v_0$ and $m$, for a set of two different inks, three nominal drop volumes, and three driving voltages.

| Ink | Nom. Drop Volume (Printhead) [pL] | Driving Voltage [V] | Drop Volume (Average, Main Drop) [pL] | $v_0$ [m/s] | $m$ [1/1000 s] |
|---|---|---|---|---|---|
| PVN | | 80 | 5.1 | 15.9 | −1.5 |
| | 10 | 100 | 5.8 | 21.5 | −1.5 |
| | | 120 | 4.4 | 25.4 | −1.6 |
| | | 80 | 25.8 | 7.7 | −0.4 |
| | 30 | 100 | 28.3 | 11.2 | −0.5 |
| | | 120 | 27.2 | 12.9 | −0.6 |
| AGF | | 80 | 31.2 | 6.0 | −0.4 |
| | 30 | 100 | 34.7 | 11.5 | −0.8 |
| | | 120 | 30.3 | 12.5 | −0.9 |
| | | 80 | 56.3 | 3.1 | −0.2 |
| | 80 | 100 | 61.4 | 5.1 | −0.1 |
| | | 120 | 63.4 | 7.2 | −0.5 |

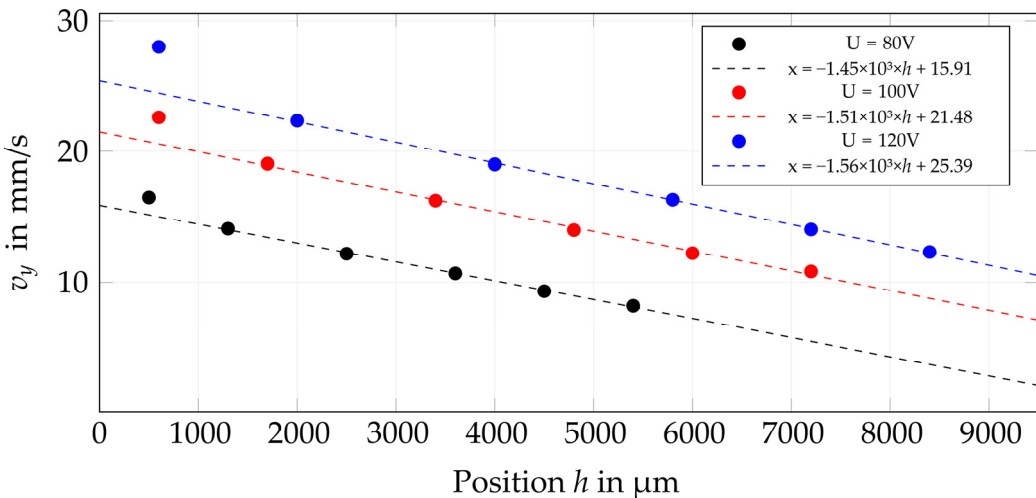

**Figure 6.** Drop speed in relation to the distance, *h*, from the nozzle plate: different linear regression curves and values for $v_0$ and *m* for an exemplary dataset [10 pL, PVN ink] for three different piezo voltages.

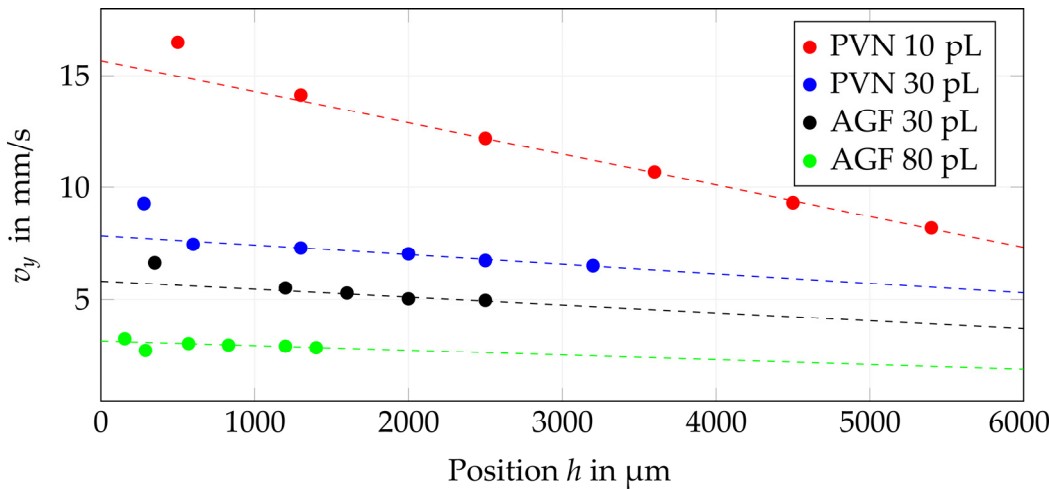

**Figure 7.** Drop speed in relation to the distance, *h*, from the nozzle plate, with a comparison between drop speed, drop volume, and ink at 80 V piezo driving voltage.

By summarizing the analysis of the drop speeds and volumes, the following conclusions can be stated:

1.  An increase in piezo voltage will increase the main drop velocity and the jetting distance;
2.  With increasing drop volume, the drop speed decreases (at a constant piezo voltage level);
3.  Piezo voltage influences the drop volume of the main drop and the development of the satellite drops.

### 5.2. Printing onto a Cylinder

Figures 8 and 9 show the results from an experiment using conductive silver ink (PVN) printed using a 10 pL printhead at 80 V driving voltage onto a cylinder with a radius of 55 mm. The procedure for getting the results will be explained.

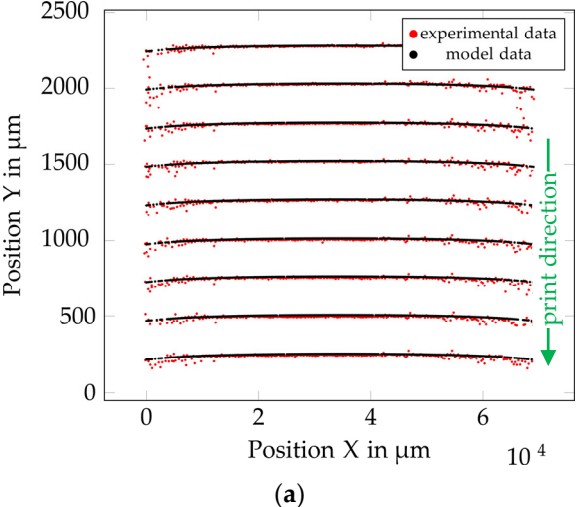
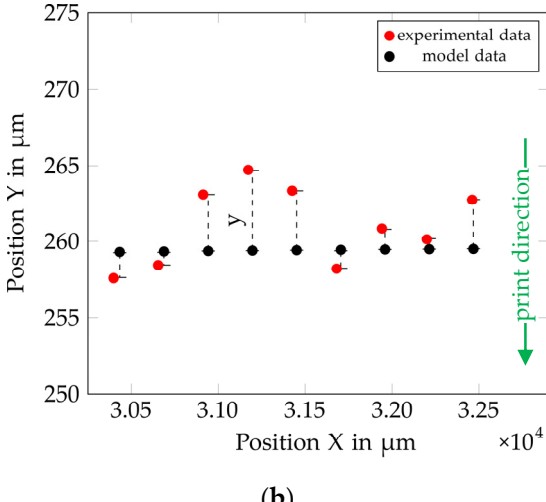

(**a**)  (**b**)

**Figure 8.** Results for PVN, 10 pL, 80 V, and $r_{a,1}$ = 55 mm. (**a**) Representation of X,Y dot co-ordinates (red—experimentally observed dot positions; black—modeled data); (**b**) magnified view of modeled and real co-ordinates of single dots and placement error, $\Delta y$, in the print direction.

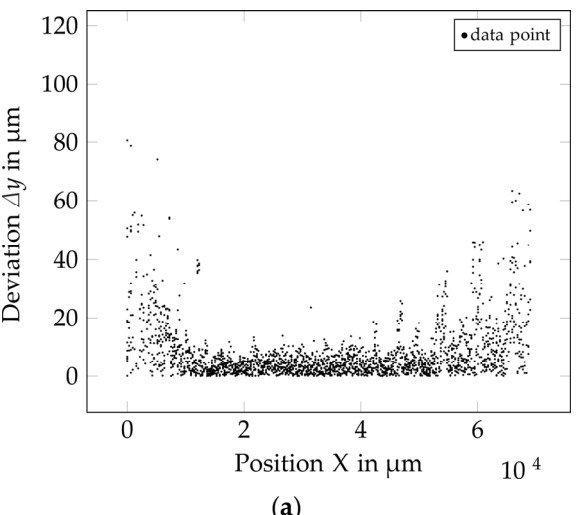
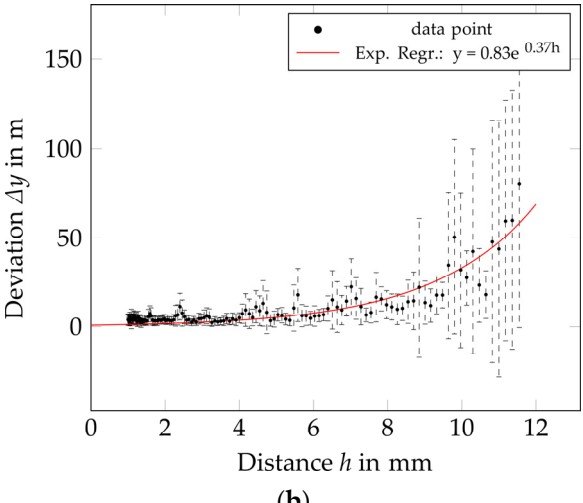

(**a**)  (**b**)

**Figure 9.** Results for PVN, 10 pL, 80 V, and $r_{a,1}$ = 55 mm; (**a**) placement error, $\Delta y$, in relation to lateral position on print out, according to cylinder curvature; (**b**) placement error, $\Delta y$, in relation to nozzle-to-surface distance, $h$.

After a row separation step to distinguish between the nine lines, the ICP algorithm was applied (line-wise) to optimally fit the model and experimental data. The experimental data (red) are shown and compared to the model data (black) in Figure 8a.

The Matlab program (see supporting information matlab files) calculated the distance, $\Delta y$, between a pair of correlated model and experimental data by using the k-nearest neighbor algorithm. Figure 8b illustrates the deviation within a magnified section of the center region of Figure 8a. The resulting deviation of all pairs was plotted against their lateral position on the photo paper in Figure 9a. In combination with the corresponding distance information, Figure 9b shows the average deviation ($n_{max}$ = 18) in relation to the distance $h$. In the center of the printout ($h \rightarrow$ min), the deviation is at a minimum. The greater the distance gets, the greater the deviation becomes.

An exponential regression function (Equation (9)) was fitted to the experimental dataset and allowed for a calculation of the deviation in relation to the distance. Table 2 gives the regression coefficients, *a* and *f*, for all investigated combinations. Additionally,

the results of the reverse calculation of the maximum distance $h_{max,i}$ (Equation (10)) are shown for a given deviation of $\Delta y = \pm 10$ μm, $\Delta y = \pm 25$ μm, and $\Delta y = \pm 50$ μm, respectively. For this dataset (PVN, 10 pL, 80 V) the resulting regression function is $\Delta y = 0.83 \cdot e^{0.37h}$. The results from Equation (10) show that at a maximum tolerable deviation of $\pm 10$ μm, the printhead should not exceed $h = 6.7$ mm of distance to the surface. At lower accuracy requirements of, e.g., $\pm 50$ μm, the maximum distance is $h_{max} = 11.1$ mm.

**Table 2.** Regression coefficients $a$ and $f$ (see Equation (9)) for exponential equation fitting and calculated maximum distance for three different deviations for all parameter sets of ink, drop volume, voltage, and radii (gray highlighted rows indicate unexpected line formations).

| Ink | Nom. Drop Volume (Printhead) [pL] | Driving Voltage [V] | $r_{a,i}$ [mm] | $a$ | $f$ | $h_{max}$ @$\Delta y = 10$ μm [mm] | $h_{max}$ @$\Delta y = 25$ μm [mm] | $h_{max}$ @$\Delta y = 50$ μm [mm] |
|---|---|---|---|---|---|---|---|---|
| AGF | 30 | 80 | 35 | 14.73 | 0.11 | −3.52 | 4.81 | 11.11 |
| | | 100 | 35 | 7.84 | 0.11 | 2.21 | 10.54 | 16.84 |
| | | 120 | 35 | 19.37 | 0.0847 | −7.81 | 3.01 | 11.20 |
| | | 80 | 45 | 10.14 | 0.14 | −0.10 | 6.45 | 11.40 |
| | | 100 | 45 | 4.17 | 0.14 | 6.25 | 12.79 | 17.74 |
| | | 120 | 45 | 8.12 | 0.0717 | 2.90 | 15.68 | 25.35 |
| | | 80 | 55 | 9.41 | 0.14 | 0.43 | 6.98 | 11.93 |
| | | 100 | 55 | 4.41 | 0.15 | 5.46 | 11.57 | 16.19 |
| | | 120 | 55 | 4.74 | 0.0805 | 9.27 | 20.66 | 29.27 |
| | 80 | 80 | 35 | 22.46 | 0.14 | −5.78 | 0.77 | 5.72 |
| | | 100 | 35 | 13.01 | 0.0909 | −2.89 | 7.19 | 14.81 |
| | | 120 | 35 | 3.05 | 0.21 | 5.65 | 10.02 | 13.32 |
| | | 80 | 45 | 11.53 | 0.16 | −0.89 | 4.84 | 9.17 |
| | | 100 | 45 | 4.62 | 0.21 | 3.68 | 8.04 | 11.34 |
| | | 120 | 45 | 1.89 | 0.21 | 7.93 | 12.30 | 15.60 |
| | | 80 | 55 | 10.1 | 0.13 | −0.08 | 6.97 | 12.30 |
| | | 100 | 55 | 9.05 | 0.18 | 0.55 | 5.65 | 9.50 |
| | | 120 | 55 | 6.44 | 0.17 | 2.59 | 7.98 | 12.06 |
| PVN | 30 | 80 | 35 | 4.85 | 0.13 | 5.57 | 12.61 | 17.95 |
| | | 100 | 35 | 13.8 | 0.0793 | −4.06 | 7.49 | 16.23 |
| | | 120 | 35 | 17.03 | 0.0586 | −9.09 | 6.55 | 18.38 |
| | | 80 | 45 | 7.94 | 0.0721 | 3.20 | 15.91 | 25.52 |
| | | 100 | 45 | 12.06 | 0.0522 | −3.59 | 13.97 | 27.24 |
| | | 120 | 45 | 15.96 | 0.0415 | −11.27 | 10.81 | 27.52 |
| | | 80 | 55 | 2.23 | 0.23 | 6.52 | 10.51 | 13.52 |
| | | 100 | 55 | 2.99 | 0.15 | 8.05 | 14.16 | 18.78 |
| | | 120 | 55 | 4.1 | 0.094 | 9.49 | 19.23 | 26.61 |
| | 10 | 80 | 35 | 3.3 | 0.21 | 5.28 | 9.64 | 12.94 |
| | | 100 | 35 | 7.07 | 0.13 | 2.67 | 9.72 | 15.05 |
| | | 120 | 35 | 9.02 | 0.0922 | 1.12 | 11.06 | 18.57 |
| | | 80 | 45 | 3.53 | 0.22 | 4.73 | 8.90 | 12.05 |
| | | 100 | 45 | 4.18 | 0.2 | 4.36 | 8.94 | 12.41 |
| | | 120 | 45 | 6.49 | 0.11 | 3.93 | 12.26 | 18.56 |
| | | 80 | 55 | 0.83 | 0.37 | 6.73 | 9.20 | 11.08 |
| | | 100 | 55 | 2.78 | 0.24 | 5.33 | 9.15 | 12.04 |
| | | 120 | 55 | 3.11 | 0.18 | 6.49 | 11.58 | 15.43 |

In contrast to the previously discussed example, other combinations showed less conclusive results. Some graphs (e.g., @ 80 pL AGF at 80 V) show huge deviations in the center or mid-range distance compared to the outer regions of the print result, where the distance, $h$, is larger. This is related to the mismatch of the applied model to the experimental data and the resulting calculation. In Figure 10a, the data for 80 pL at 80 V printed on a cylinder with $r_{a,1} = 55$ mm show an unexpected distribution of dots. The printed lines appear like a riders bow, where, in the center area, the line is bent in one direction, and toward the outer region, this alters the bend direction. The resulting average

deviation in relation to the distance can be observed in the graph in Figure 10b and could lead to false conclusions. An increase of the piezo voltage to 100 V and 120 V (volume 80 pL, $r_{a,1}$ = 55 mm) saw the dot distribution on the surface improve, and the discrepancy of the experimental data to the model was decreased. Out of the 36 combinations, 17 were not fitted ideally to the calculated model due to unknown effects, which led to an unequal distribution of the dots across the line, as shown in the example in Figure 10. For the 80 pL AGF dataset, 8 out of the 9 combinations showed this unexpected behavior. The data for 4 out of the 9 combinations did not perfectly match the model for the 30 pL drop volume for both inks (PVN and AGF). Just one outlier was observed for PVN at a drop volume of 10 pL. In Table 2, the outliers are marked with a gray cell background. One explanation for the false prediction of the drop positions on the cylinders might be related to a higher volume and, thus, lower speed when compared to the other combinations for 10 pL and 30 pL, respectively. Air turbulence arising from the printhead motion itself might also influence the drop trajectory of slower and bigger drops to a greater extent.

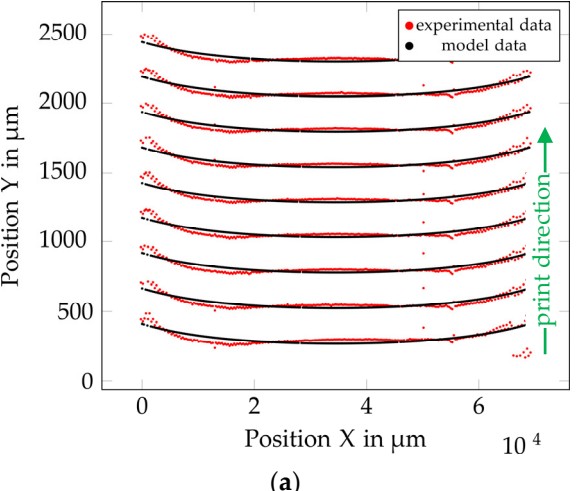

(**a**)

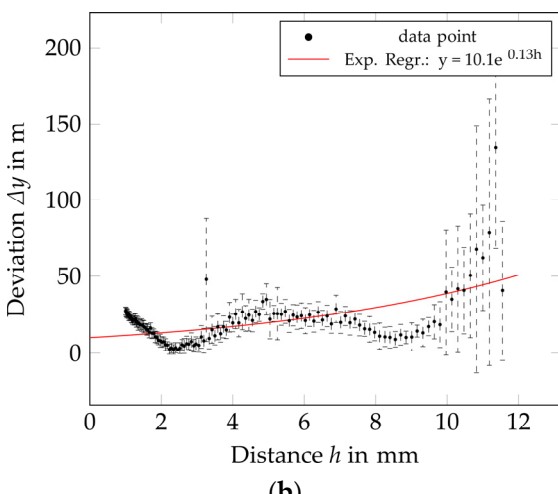

(**b**)

**Figure 10.** Example of model vs. experimental data for 80 pL at 80 V driving voltage and $r_{a,1}$ = 55 mm: (**a**) position of dots on photo paper; (**b**) deviation in relation to surface-to-printhead distance with regression curve.

## 6. Conclusions and Outlook

The investigations discussed in this paper represent fundamental experiments that can help to understand the achievable accuracy of robot-controlled inkjet printing onto 3D objects. By observing different positions from the nozzle plate, the development of the drop speed and the influence of the piezo voltage and drop volume were determined for a conductive silver and an insulating UV ink. The results of extensive printing experiments on cylinder targets showed that the deviation of droplets from their ideal position (according to the model) is highly dependent on the distance between the nozzle and the surface. Furthermore, the tuning of printing parameters (piezo voltage) allows for an increase in the throw distance and achievable accuracy. For the different parameter sets, a method was described to calculate either the resulting placement deviation in relation to the distance or vice versa, allowing us to calculate the distance limits.

The comprehensive investigations showed that the prediction of droplet positions regarding cylinder geometries is difficult. The unexpected appearance of printed vertical lines for a number of parameter sets was observed. For slow (<16 m/s) and big drops (30 pL and 80 pL), the resulting line shape especially differed from the modeled lines. Within the investigated parameters, the silver ink printed with a 10 pL printhead (nom. drop volume) achieved the best match to the model and also showed the highest drop speed.

While a robot-guided inkjet printing process is able to perform printing in all room directions, future research should address the influence of the spatial orientation of the

printhead on drop trajectory and placement accuracy. For this case, the proposed method can be used by simply rotating the cylinder to another orientation to realize sideways or bottom-up shooting. In addition, the effect of geometric distortion and position accuracy on layer formation in coherent functional layers is of great importance and should also be investigated.

This research is a contribution toward the improvement of the robot-guided inkjet printing approach in terms of productivity and the efficient determination of drop placement accuracy. The benefit of a cylinder setup is a line of nozzles ejecting drops under the same working condition while allowing for different distances to the print surface. With this setup, the placement accuracy of one parameter set in relation to the jetting distance can be investigated with one single print run. In combination with automated image acquisition and processing, the proposed method can dramatically decrease the effort needed to determine the drop placement error of different sets of print settings (printhead, drop volume, ink, driving voltage, and movement speed). The implementation of the information regarding deviations in jetting distance in relation to surface curvature for image preparation allows the user to compensate for image distortion and improve print and functional layer quality.

**Supplementary Materials:** The following supporting information can be downloaded at: https://www.mdpi.com/article/10.3390/machines11050568/s1, PDFS1: Diagrams_Deviation_dY_vs_distance _h_with_exponential_fit.pdf; PDFS2: Diagrams_Deviation_dY_vs_X_position.pdf; PDFS3: Diagrams_Dot_Coordinates_vs_Model_Data.pdf; 5 Matlab files: posmodel.m, pairNN.m, run_me.m, splitNres.m, icp.m with example dataset coordinates_PVN_10pL_80V_55mm.csv for calculation of dot deviation for one parameter set.

**Author Contributions:** Conceptualization, R.T.; Methodology and Investigation R.T. and D.M.; Writing—original draft, R.T., A.W. and D.M.; Writing—review & editing, R.T., A.W. and D.M.; Supervision, A.W. and R.Z. All authors have read and agreed to the published version of the manuscript.

**Funding:** This research was funded within the framework of AiF/IGF (project name: "3D Robojet"; IGF-funding reference: 20606 N; runtime: 01.03.2019–31.07.2021) by the German Federal Ministry for Economic Affairs and Energy (BMWi).

**Data Availability Statement:** The data presented in this study are available upon request from the corresponding author.

**Acknowledgments:** I would like to thank my supervisors Andreas Willert and Ralf Zichner for helping and encouraging me to create and finalize this publication.

**Conflicts of Interest:** The authors declare no conflict of interest.

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
