# Peer review of "Novel and Efficient Methodology for Drop Placement Accuracy Testing of Robot-Guided Inkjet Printing onto 3D Objects"

_machines, doi:10.3390/machines11050568_

Round 1

Reviewer 1 Report

Author mentioned " In summary, all factors described in section 2 have to be considered" in line 182. But " Liquid Flow Behavior" was never talked about in a later experiment/result/discussion session. Is it negligible? If so, please also mention it.    Looks like the paper is not complete yet.  "Printhead Spatial Orientation" experiment was not in the context. Expecting to see one data with print head pointing up and another one pointing sideways. If the experiment can not be done in the current setup, please indicate the challenges (print head limitation etc. ) and mention what to be solved in future studies.    Figure 8 (a) is not consistent with Figure 10 (a), please include an arrow indicating the motion direction of the print head in the figure.    In Figure 8 and Figure 10, Please include subfigure (c) with 9 shots of droplet with print head stay still. By comparing (c) and (a) we know how much print head movement affects the data.    Please mention the nozzle diameter/other difference between 30pL nozzle head and 80 pL nozzle head. Most readers may have no idea what it means by saying XX pL nozzle head. 

Reviewer 2 Report

It should be further elaborated what exactly is so outstanding about this new approach. The work will be of interest to a small circle of researchers, perhaps the focus can be expanded? Your work has become more of an interesting reference work. The results are explained and comprehensible. However, are the results important and groundbreaking? Probably not for the broader audience, but a few applications can already benefit from them. The elevated data is reliable, the methods are clearly evident, and the conclusions are easily understandable. But it is not an outstanding and exceptional piece of work.

The English is okay, although sometimes a bit colloquial.

Reviewer 3 Report

Thalheim et al., submitted the manuscript entitled “Novel and efficient methodology for drop placement accuracy testing of robot guided inkjet printing onto 3D objects” to “Micromachines (I. F = 3.523)”. This is an interesting paper on robot guided 3D printing technology. It can be published after minor revision.

1. Introduction looks fine, cite the following references for 3D printing technology; 1. Nat. Rev. Mater., 2018, 3, 84 -100, 2. Adv. Funct. Mater. 2021, 31, 2102777 and 3. Polymers 2021, 13, 598.

2. For many proposed equations, explanation for some symbols and denotations must be supplemented in discussion.

3. Figure 8, 9 and 10 are looks great in the printed papers, please enhance the resolution of Figures and inset font text.

4. Explain the use of this 3D printing technology if the driving voltage more than 120 V (Note: growing countries such as India use 240 V DC), so it is essential to justify this technology at greater than 120 V.

5. Concise the conclusion section with out affecting the outlooks. Also, update the literature with recent papers.

English looks fine, need to check spelling in many sections

Round 2

Reviewer 1 Report

Please add  " print direction" and green arrow for figure 8 as well. 

Please upload word version with track change or pdf with final version. I see some figures have labels crossed out. (Figure 4 (b) and Figure 7 etc). Please correct them. 

 English language fine. 
